

# One second vector and scalar magnetic measurements at low latitude observatory, CPL

Phani Chandrasekhar Nelapatla[1]., Sai Vijay Kumar Potharaju[1]., Kusumita Arora[1]., Chandra Shakar Rao Kasuba[1]., Leonid Rakhlin[2]., Sergey Tymoshyn[2]., Laszlo Merenyi[3]., Anusha Chilukuri[1]., Jayashree Bulusu[1]., Sergey Khomutov[4]

[1]Hyderabad Magnetic Observatory, CSIR-National Geophysical Research Institute, Habsiguda, Uppal road, Hyderabad-500007, India

[2]Research Centre GEOMAGNET, Pavlika St.,79005 Lviv, Ukraine

[3]Geological and Geophysical Institute of Hungary, H-1143 Budapest, Stefániautca 14

[4]Institute of Cosmophysical Research and Radio Wave Propagation, FEB RAS, Kamchatka State Technical University, Russia.

*Correspondence to*: Phani Chandrasekhar (phaninelapatla@gmail.com)

**Abstract**

There is increasing demand from the global geomagnetic community for the recording of 1 second vector and scalar magnetic data in lieu of the traditional of the 1 minute data, as the 1 second magnetic data would be more compatible with measurements made from low-earth orbiting satellites and the increased detectability threshold, would contribute to: (i) understanding the global scale ultra-low frequency (ULF) waves, sudden impulses and other processes in the ionosphere & magnetosphere: (ii) development of real-time space weather forecasts. The combination of ground and satellite data opens a new pathway in understanding many underlying physical processes in the lower-middle atmospheric dynamics, which has not been accurately understood so far.  The International Real-time Magnetic Observatory Network (INTERMAGNET) observatories (IMO-s) have taken a lead in this direction and many IMO-s now produce both 1 minute and 1 second data. Being affordable, rugged and compact as well as having low power consumption, fluxgate magnetometers are the staple vector sensors of IMO-s. The increased order of noise in these sensors with increasing frequencies, is the main concern and work has been going on for the last decade towards development of suitable instruments (Courtillot and Chulliat, 2008; Korepanov et al. 2006, 2009; Pedersen and Merenyi, 2016 and references therein, Dobrodnyak, 2014; Logvinov, 2014) and techniques for the evaluation and elimination of noise from the data is also being pursued (Turbitt et al. 2013).

At the new Magnetic Observatory of CSIR-NGRI in Choutuppal (CPL) campus, 1 second magnetic measurements commenced in the year 2016 using the newly developed Observatory grade 1 second fluxgate magnetometer, GEOMAG-



02MO, from GEOMAGNET Ukraine and the Overhauser Proton Precession Magnetometer along with the data acquisition system, MAGREC-4B. The processes of commissioning of this setup in low-latitude conditions, with the aim to finally produce 1 second definitive data (the standards of which are still under discussion with INTERMAGNET) and the characteristics of the data from this new instrument are presented in this work.

Keywords: One second data, Fluxgate magnetometers, Overhauser, GEOMAG-02MO and MAGREC-4B.

## 1 Introduction

Across the world, 200 Magnetic Observatories are in operation, of which 150 are INTERMAGNET Observatories (IMO-s) recording high quality 1 minute vector and scalar data. Many of the IMO's started recording and producing 1 sec data, prominent ones being 11 Magnetic Observatories operating by Institut De Physique Du Globe De Paris (IPGP) (Courtillot and Chulliat, 2008);3 Polish observatories maintained by Institute of Geophysics Polish Academy of Sciences (Reda and Neska, 2016); 4 observatories operating by the Japan Meteorological Agency (JMA) (Minamoto, 2013); 8 observatories of

British Geological Survey (BGS) (Thompson, 2014), and some others operated by different academic and research institutes (see for more information, http://www.intermagnet.org). Data from Magnetic Observatories often in combination with satellite magnetic data, is widely used for scientific applications, such as  (i) calculating time-varying core field models, (Hulot et al. 2007; Jackson and Finlay, 2007); (ii) studying rapid processes in the core, like geomagnetic jerks (Courtillot et al., 1978; Holme, 2007); (iii) studying the various electrical current systems in the Earth's ionosphere and magnetosphere,

both during short events such as magnetic storms, sub-storms, sudden impulses and Pi2 pulsations  (Kennel, 1996; Russell, 2005) and also on longer time scales (McPherron, 2009).

While on one hand we are living in increasing technology driven society, where the very pathways of high speed communication is affected by the changes of the magnetic field and space weather conditions, on the other hand, the very

spread of human habitation and technology is an impediment for measuring changes of the Earth's magnetic field devoid of anthropogenic influences, rendered particularly difficult for the more populous countries. Apart from the availability of suitable location and size of space for a magnetic observatory, access to infrastructural facilities viz. trained man power, uninterrupted power supply, non-magnetic & high thermal insulated materials for the construction of magnetometer housing, high speed internet for the transmission of data in real-time to headquarters are equally important for the long time recording

of quality data at the observatories. Further, the allocation of funds for setting up and maintaining observatories is often a constant challenge. The newly established 1 second system at the CPL Magnetic Observatory may serve to be an example of low cost, rugged installation with data generation of sufficient quality in high temperature, low latitude regions. The following sections elucidate the experimental setup for the commencement of one sec magnetic variations, system description of GEOMAG-02MO and software package, MAGREC-4B data acquisition, real-time data transmission from



CPL to HYB with a latency period of 2 minutes, raw and filtered data sample from GEOMAG-02MO and GSM-90F1 through MAGREC-4B, despiking of data using different cut-off frequencies by filter software and the observed effects of temperature on the dataset and treated.

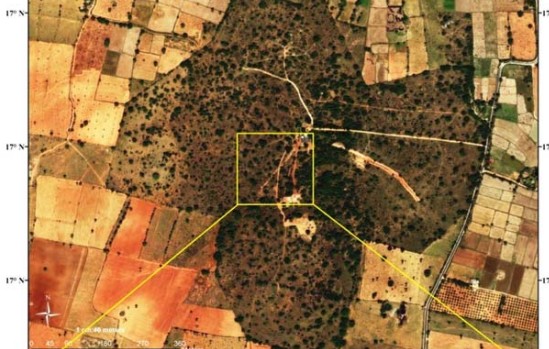

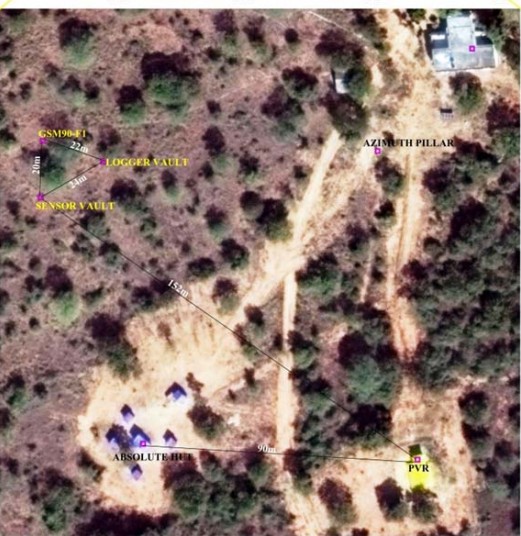

Figure 1: Bird's eye view of Choutuppal campus of CSIR-National Geophysical Research Institute (top panel) and the highlighted text box show the location of variometer vaults, Azimuth pillar, Absolute hut and Primary variometer room (PVR) of Choutuppal Magnetic Observatory (bottom panel).

20  The CPL Observatory is 65 km away from Hyderabad (HYB) due east and along the same latitude and developed as an alternate/addition to the IMO-HYB (Arora et al. 2016). Figure 1 shows the layout of the Observatory complex: the Primary





Variometer Room (PVR), which houses the 3-component FGE Magnetometer from DTU and FGE along with Overhauser and the Absolute Hut for the D-I measurements; about 90 m to the west. The setup of the new fluxgate is confined to a couple of shallow vaults, approximately 20 m apart, about 150 m north west of the PVR, where the recently developed Observatory grade one second magnetometer, GEOMAG-02MO manufactured by the Ukrainian company GEOMAGNET,

has been installed and is being tested for data quality, stability, consistency.

Different fluxgate magnetometers are in operation for recording 1 second 3-component magnetic field variations at magnetic observatories worldwide; the most widely used ones being, 3-component FGE Magnetometer by Danish Technical University (DTU), LEMI by Lviv Centre of Institute for Space Research, Vector Magnetometer-391 by Institut de Physique

du Globe de Paris and GEOMAG-02M by GEOMAGNET. The standard for scalar field measurements is the GEM Systems GSM-90 Proton precession magnetometer (Overhauser magnetometer). The previous version of fluxgate by GEOMAGNET, the GEOMAG-02M is an unsuspended sensor with compact electronics having very low power requirements, which makes it suitable for many observatories as well as remote variometer sites. The main advantages of using GEOMAG-02M and its noise characteristics are extensively discussed in Jan Reda and Mariusz Neska (2016). The upgraded version of this

magnetometer, GEOMAG-02MO is now available to the community following the standards of INTERMAGNET 1 second data requirements and the details of the system are discussed in this paper.

## 2 Experimental setup & System Description

### 20   2.1 Design of the setup

Two non-magnetic vaults of dimension 4′ x 4′ x 4′ were constructed at CPL Observatory campus for deploying the 3 component sensor and house the data logger systems (Figure 2). Extruded Polystyrene (EPS) foam sheets of 4 inch thickness of dimensions 4 ′ x 8 ′ are used as thermal insulator material inside the four corners of the vaults to achieve good temperature

control both for the sensor and electronic units. Utmost care was taken for controlling temperature variations in the sensor and logger vault, as GEOMAG-02MO is having a typical temperature coefficient of ~0.2 nT/°C. Therefore to have a good quality of measurements free from temperature effects, two different sizes of enclosures were made and kept on the top of sensor, to make sure that the temperature variations should be almost constant over the day (Figure 2). On the top of these enclosures, EPS sheets were spread over to close the outer lid of the sensor vault. The sensor vault is protected by a

wooden door layered again with the EPS sheets on the top, over which 7′ x 7 ′ white marble was kept to achieve a good temperature control for the sensor. Similar arrangements were made to the data logger vault (Figure 2). The sensor-logger vaults and the GSM90-F1pillar are separated by about 20 m, almost equidistant from each other. The vector and scalar magnetometers are connected to their respective GPS receivers, enclosed in a polyvinyl chloride (PVC) pipe, provide the details of the accurate time stamping for field variations, geographic co-ordinates and elevation of the site (Figure 2).

Lightning arrester is mounted on the top of the data logger vault and the other end of it is grounded deep into the earth to

protect the electronics of the recording systems during severe thunderstorms, which happens during the monsoon. Uninterrupted power supply was provided for the recording systems as well as to the computers using 3 kW solar panels.

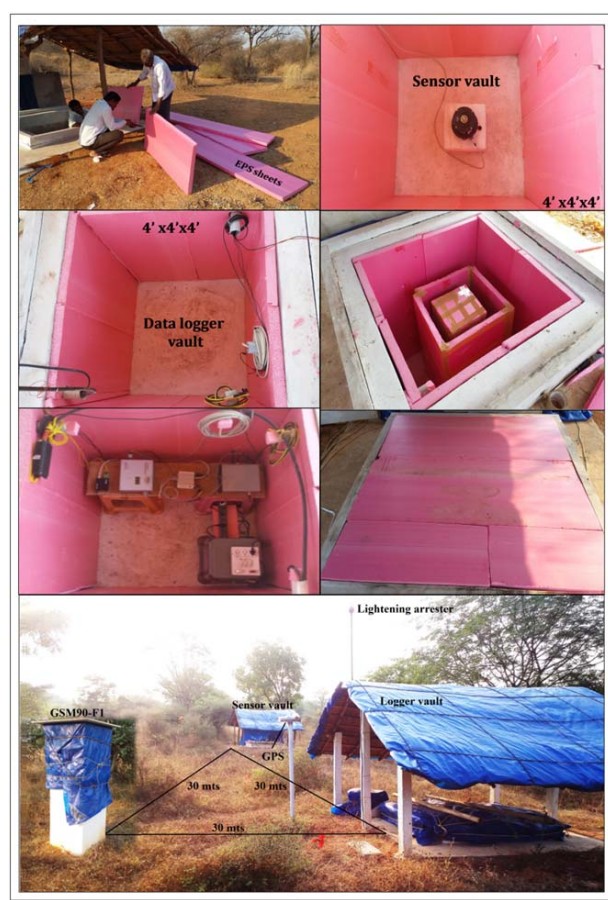

Figure 2: Thermally insulated new variometer vaults using Extruded Polystyrene (EPS) foam sheets for good temperature control and the location of sensor and logger vaults.

GEOMAG-02MO and GSM90-F1 are installed on non-magnetic pillars as shown in Figures 3a and b. GEOMAG-02MO sensor is oriented nearest to the magnetic north leaving 'Y 'in compensated mode, GSM90-F1 sensor is oriented in East-West direction. GEOMAG-02MO comes with automatic compensation of the external field, which is sequentially adjusted. GEOMAG-02MO and GSM90-F1 sensors are connected to the respective data loggers via undergrounded connecting cables of length 30 m. The two data acquisition systems are further connected to MAGREC-4B data acquisition system via RS-232 cables (Figure 3c). More information about the configuration of MAGREC-4B to the experimental setup and the data



transmission techniques in real-time from CPL to HYB Magnetic Observatory is discussed in the next section. Optical Fiber cable of length 110 m was laid underground between the MAGREC-4B data acquisition system located in logger vault to the main building of CPL Observatory, and connected to a computer to visualize field variations in real-time as well as to store the data in hard drive of the Linux computer (Figure 3d). Pier corrections are regularly being carried out using an additional

5 GSM-19W scalar magnetometer apart from the GSM90-F1, to make sure the established spatial gradient is constant during

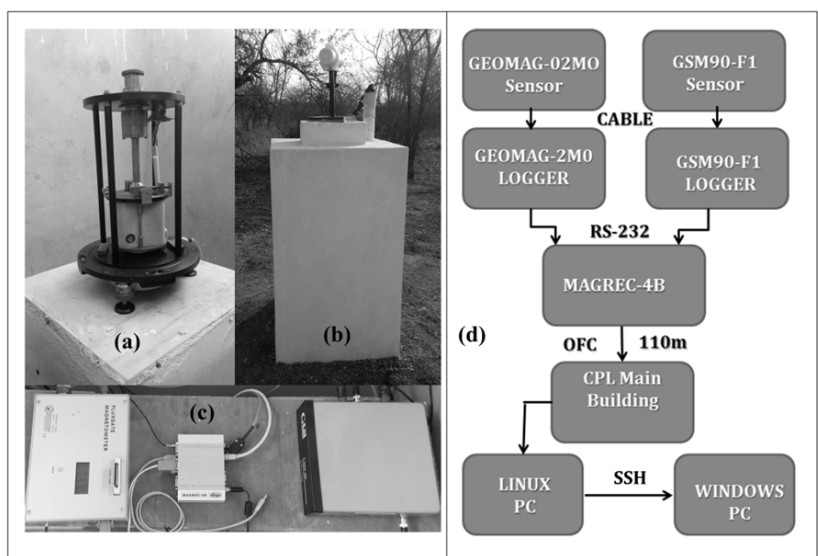

Figure 3: (a) Three component GEOMAG02-MO (b) GSM90-F1 (c) Data loggers of GEOMAG-02MO, GSM90-F1 and MAGREC-4B data acquisition system and (d) details of connections used for the experimental setup.

the days of observations. One second raw data in real-time is transmitted to CPL main building from MAGREC-4B and with
10 a latency period of 2 minutes, the data is transmitted to HYB Observatory through internet from a Linux machine. The daily data after adding the calculated latest baseline measurements is investigated the next working day at HYB Magnetic Observatory for preliminary processing like: despiking and removal of disturbances from artificial origin. MATLAB based codes are being used for processing the daily data and intended to extend the codes for preparation of quasi and definitive data preparation.

## 2.2 GEOMAG-02MO instrument and Software

At creation of the magnetometer GEOMAG-02MO for observation in the conditions of the observatory, conforming INTERMAGNET Definitive one-second Data Standard dt. 07.09.2012, the main attention was focused on the following
problems:



- Reduction of the noise level,

- Digital data filtering,

- Reduction of non-orthogonality of components of the MT sensor.

5    Optimization of sensor excitation while increasing its field excitation significantly improves noise characteristics. This
instrument achieves the noise density 10 pT at a frequency 0.1 Hz (Figure 4a). To fulfil the requirements regarding the
temporal characteristics (Time-stamp accuracy: 0.01s and Phase response:  Maximum group delay: ± 0.01s) a program of
digital filtering of measured data has been introduced in the computer after receiving them via the RS-232 port. The program
allows the choice of cut-off values and the width of the window. Thus, to suppress the high level of interference caused by,
10    for example, the railway, the frequency of sampling analog-to-digital converters in the measurement channels increases to
100 samples per second with subsequent filtering and averaging. Figure 4b shows a graph of noise density in the frequency
range from 8 mHz to 0.2 Hz as a result of the low-frequency digital filtering at cut-off frequency 0.2 Hz, the width of the
window 25 seconds. Constraints in technical specifications in the production of components of the magnetic field sensor and
its housing allow to achieve non-orthogonality of the axes not better than +/- 30 arc minutes. During long term operation this
15    value remains stable due to the use of artificial glass (SITAL) or marble that have a high stability in a wide temperature
range and time. Further reduction of non-orthogonality is achieved by using digital compensation. Method of compensation
allows reduction of the specified error up to several units' arc minutes. Specifications are given in Table 1.

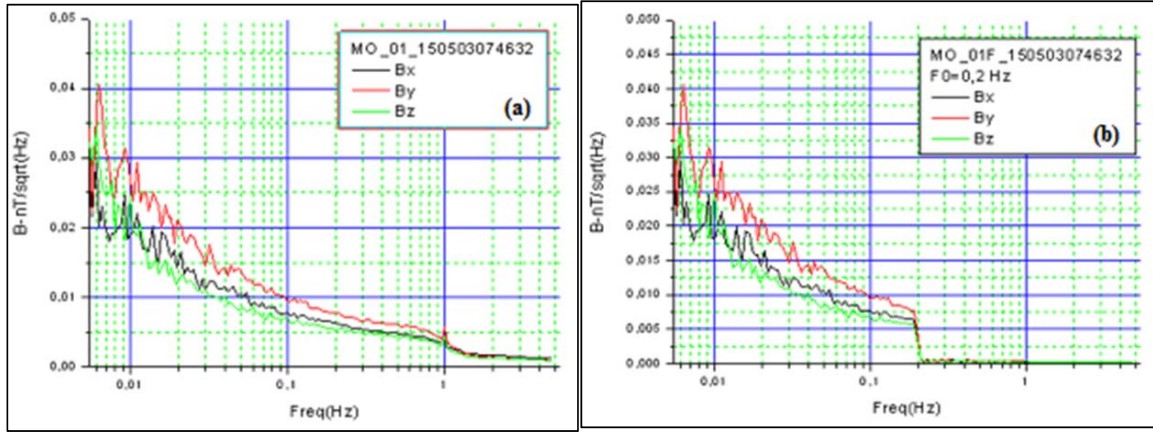

Figure 4: (a) Noise characteristics of different components of GEOMAG-02MO. (b) Noise characteristics as a result of
digital filtering at F0 = 0.2 Hz.

25



| Main technical characteristics of  GEOMAG-02MO | |
|---|---|
| Measuring range of full magnetic field (MF) | ± 65000 nT |
| Measuring range of MF variations | ± 4000 nT |
| Resolution of MF variation resolution | 0.001 nT |
| Temperature drift | <0.2 nT/°C |
| Tolerance of component non-orthogonality of magnetic field sensor | <30 ang. min |
| Non-orthogonality error sensor component MF after compensation | <5 ang.. min |
| Automatic compensation range of constant MF each component | ±65000 nT |
| Measuring channel information sampling number | 100 1/sec |
| Operating temperature rage | - 10 to +40° |
| Connecting cable length between the sensor and electronic unit | Up to 50 m |
| Power consumption | 12 V; 0.1 A |
| Capacity of Flash card | 64 MB to 64 GB |
| Data transfer & online magnetogram through Logger software via Rs 232 to USB | Windows platform |
| GPS timing and co-ordinates determination with elevation | 0.01s accuracy |
| Server connection from Desktop machine via Logger software | Internet connection |

Table 1. Technical specifications of GEOMAG-02MO

GEOMAG-02MO comes with Windows platform based Low Pass Filter (LPF) GEOMAG software for despiking the noise in the data and the software is developed based on the 1 second data standards recommended by INTERMAGNET council

10 (Turbitt et al. 2013), Figure 8. The LPF GEOMAG performs the following processing functions: (i) 3-channel data processing via a digital SINC– low pass filter; (ii) Data averaging over a pre-determined time interval (i.e. from 5 sec to 60 seconds); (iii) Frequency cutoff: 0.001 to 0.990 Hz and Filter window: 1.0 to 49.0 Sec; (iv) Filter types: Hannah, Hamming and Blackman; (v) Data output format: 1 pT and 10 pT. After the filtering process is finished, the LPF GEOMAG window shows the report of generated files: (i) Filtered data and (ii) Noise data. For the present experimental setup at CPL, the LPF

15 GEOMAG software was configured to the 1 second data from MAGREC-4B data acquisition system, for despiking the vector and scalar magnetic data. Description of the MAGREC-4B data acquisition system and the design of the variometer vault are discussed in the next section. The averaging feature for LPF GEOMAG software is under progress for the data output from MAGREC-4B data acquisition system.



GEOMAG-02MO also offers two additional Windows platform based software's: (i) the GEOMAGNET (GM) Logger and (ii) GM Logger server software. The GM Logger software is intended to receive the information from the fluxgate magnetometer by plotting the real-time field variations and saves the data to hard disk of computer via RS-232 to USB connector. The GM Logger server software: receives information files from several FTP servers, uncompress the received

files on specified folders on computer, saves information of log files on local database, shows information of received files in calendar view and also can concatenate several succession log files to one.



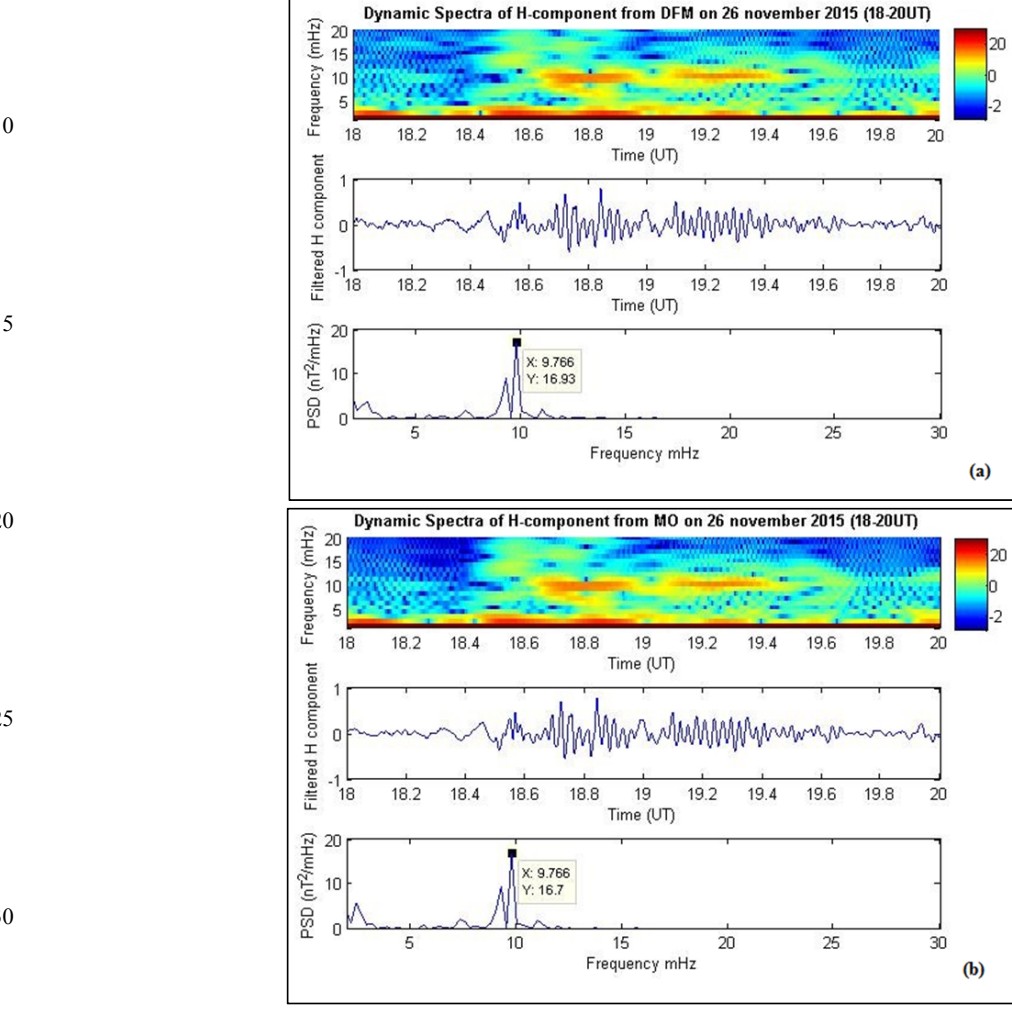

Figure 5: (a) Spectral decomposition of Quiet day night time data for FGE on 26 Nov 2015 (b) Spectral decomposition of
Quiet day night time data for MO on 26 Nov 2015



Before installation in CPL, the MO was installed at HYB and the data was compared with that from the FGE, which has been generating definitive data over the last decade. 1 second night time H component data for a Quiet Day on 26 Nov 2015, for each instrument is compared by analyzing the spectral content of the data; Figure 5a and b show the similarity of

response.

**2.3 Data acquisition system: MAGREC-4B**

Magrec-4B is a Dual-Core Intel® Atom™ processor based small-size fan-less data acquisition computer equipped with
PalmAcq GPS timing module. Magrec-4B runs a middleweight Linux operating system, containing the MAGLIN data acquisition program and several related utilities. Magrec-4B is primarily designed for geomagnetic Observatory data acquisition (Figure 3c). However, it can be used for many other data acquisition, data monitoring purposes. Magrec-4B supports long-term, unattended data acquisition from one or more instruments. For more information on technical specifications of MAGREC-4B, see http://www.mingeo.com/prod-magrec4b.html. The MAGREC-4B screen, which
receives data from GEOMAG-02MO and GSM90-F1 systems is illustrated in Figure 6. Figure 6a show the window (F1), which provides the details of the Observatory and the operating instruments for recording the magnetic field variations. This window also offers information about the log report, i.e. time of initiation of magnetic field measurements, sampling interval, time stamping error and restart time of logger systems. This information will be stored on monthly basis and available to the user at any time to know the complete details of the recording systems. MAGREC-4B offers multiple
plotting tools for visualization of real-time data: (i) 1 sec raw 3 component data from GEOMAG-02MO (Figure 6b); (ii) 1 second raw vector and F data from GEOMAG-02MO and GSM90-F1 (Figure 6c); (iii) 1 minute averaged INTERMAGNET Gaussian filter applied for 1 second data for GEOMAG-02MO and GSM90-F1 (Figure 6d); (iv) raw supplementary data of sensor and logger temperatures of GEOMAG-02MO (Figure 6e); (v) minute supplementary data of sensor and logger temperatures of GEOMAG-02MO (Figure 6f); (vi) raw data from GSM90-F1 (Figure 6g); (vii) 5 sec temperature data of the
MAGREC-4B data acquisition system (Figure 6h); (viii) 10 minute mean temperature data of the MAGREC-4B data acquisition system (Figure 6i); (ix) raw GPS data from MAGREC-4B data acquisition system (Figure 6j); 10 minute mean GPS data from MAGREC-4B data acquisition system (Figure 6k); the details of the sampling interval, field variations from GEOMAG-02MO as well as from GSM90-F1 are available in windows (F3 and F4, not shown  in the figure) and (x) the CPU temperature of the MAGREC-4B data logger and the sampling interval is available in window F5 (Figure 6l).






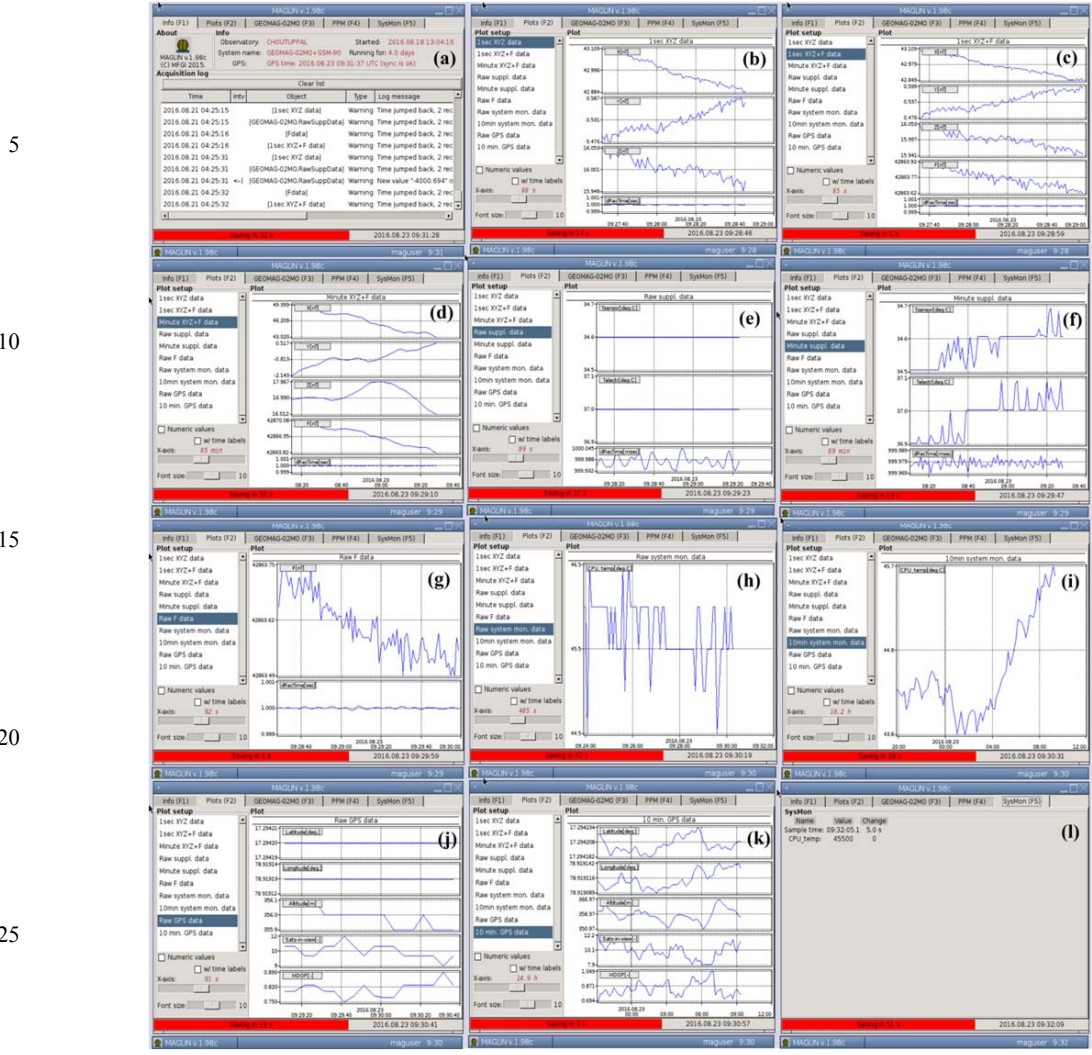

Figure 6: Details of data logging and real-time plotting tools of MAGREC-4B data acquisition system.

Magrec-4B comes with pre-installed Debian Linux and has a total of 4GB Hard drive, which somewhat limits the data storage and if there is any hard disk failure, there is possibility of loss of data as well. In order to keep the data secured we have configured a computer on Linux platform, which communicates with Magrec-4B via internet/lan so that the data once received from the instruments to Magrec-4B will simultaneously be transferred to this computer automatically (Figure 3d). A



new script is installed in Magrec-4B to perform this process. Whenever there is a network dis-connectivity between Magrec-4B and the Linux machine, the script transfers the data via lan and when the connection is re-established the script rechecks the data which was last transmitted and again sends the data from where it was stopped, to make sure the data is secured without any loss at the main control room at Choutuppal. The transmission of the data to Hyderabad has to be completed in

another stage.

**2.4 Near Real-time data transmission from CPL to HYB Magnetic Observatory**

As the setup in Choutuppal is unmanned, it is essential that the data be transmitted in near real time to the main

establishment in Hyderabad, where a Windows machine was configured with a Public IP to receive data from CPL Linux machine (Figure 3d). Internet connectivity has been achieved for the Observatory by laying 2.5 km of OFC cable from the Choutuppal town to CPL Magnetic Observatory; currently speed of transmission is found to be sufficient as data volume is low, however there are occasional phases of interruption. In effect, the data is stored in the CPL Linux computer and also transmitted to a computer in Hyderabad. The computer at Hyderabad is on Windows platform, configured with WINSCP and

related .exe files which checks the continuity of data flow from CPL and in cases of interruption, re-writes the data once connection is re-established. This will make the data secured at CPL Linux machine as well as HYB Observatory.  Any further data transmission to any corner of the world can be easily done from HYB Windows machine. The parameters of the data transmission are further explained in detail in Figure 7.

```
C:\PROGRA~1\WinSCP>winscp.com /script=synctolocal.txt
batch           abort
reconnecttime   120
confirm         off
Searching for host...
Connecting to host...
Authenticating...
Using username "root".
Authenticating with pre-entered password.
Authenticated.
Starting the session...
Session started.
Active session: [1] root@103.242.153.146
Comparing...
Local 'D:/data' <= Remote '/root/magrec'
Synchronizing...
Local 'D:/data' <= Remote '/root/magrec'
201608.sysmon-10min.dat  :       2 KB :    0.0 KB/s : binary : 100%
20160819.dat             :     652 KB :  135.1 KB/s : binary : 100%
20160819.F               :     341 KB :  167.0 KB/s : binary : 100%
20160819.gps.raw.dat     :      30 KB :  161.0 KB/s : binary : 100%
20160819.scc             :      72 KB :  146.5 KB/s : binary : 100%
AUG1916.MIN              :      14 KB :  133.6 KB/s : binary : 100%
AUG1916.SUP              :       5 KB :  127.4 KB/s : binary : 100%
cpl20160819vmin.min      :      19 KB :  124.3 KB/s : binary : 100%
cpl20160819vsec.sec      :     255 KB :  135.2 KB/s : binary :  23%
```

Figure 7: Details of real-time data transmission from Host to Client machines using secured algorithms and scripts



We have used a Batch file with "Abort" option and Confirm with "Off" option so as to make sure the connection at client end is with repeated rechecks for a default time limit of 120 seconds. The session starts by checking the Host ID username

and the authenticated pre-entered password with RSA (Rivest, Shamir, and Adelman) Key through SFTP (Secured File Transfer Protocol). The terms 'Comparing' and 'Synchronizing' in the figure shows the details of the data transmission from host to client machine at regular intervals with a time interval of 120 seconds. From the MAGREC-4B data acquisition system, we have selected 9 data parameters of binary files as shown in the Figure 8 for transmission of real time data to the Client machine. The details of the file size, of each data parameter as well as the speed at which the data is being

transmitted from the host to the client machine are also shown in the same figure. The percentages shown in column 5 of Figure 7 show the data transmission and updating process at client machine. 100% data transfer is achieved only when 1 sec data is copied with the latest records of 120 seconds and also the client machine rechecks the data by synchronizing the earlier records of the current day. The example of on-going process of data transmission with the latest records and the updating process is also shown in the same figure at Row 9. Once the data is synchronized with the latest records the 23% of

file transmission (each) will become 100% on completion of this task, with further synchronizing with the earlier saved data. The file size of the above said 9 parameters keeps increasing for every 120 seconds of the data being updated at the host machine. The whole process is repeated for each cycle of 120 seconds till the day is completed.

**3 Data**

**3.1 Raw and Filtered data sample from GEOMAG-02MO and GSM90 through MAGREC-4B**

Samples of 1 second raw data from CPL MO system is shown in Figure 9, H (Figure 9a), D (Figure 9b), Z (Figure 9c) and F (Figure 9d) components for one day, i.e. 28[th] January 2017. There were few noticeable spikes in the data set, which are

despiked using LPF GEOMAG software using the filtering parameters as shown in Figure 8. The top panel of Figure 8 illustrates the loading of 1 second data from MAGREC-4B into LPF GEOMAG software. This header file contains the complete information about the format of the data, instruments deployed, name and co-ordinates of the Observatory, reporting magnetic field components, sensor orientation, sampling interval, data type and the field variations of the day. Once the raw data file is uploaded into the LPF GEOMAG software, the information about the filter parameters will be

available in the same file, as shown in the figure. The information about the filter parameters is shown in the middle panel of Figure 8. The available filter parameters are already explained in earlier sections. For despiking, the cut-off frequency is set to 0.005 Hz with Blackman filter window 25.0 s. The filtering is enabled for all the components and the output format of the data is 10 pT. After running the LPF GEOMAG filter, the software generates three output data files: (i) the raw file; (ii) filtered file and (iii) the noise data file. The filtered data using LPF GEOMAG software for the day 28[th] January 2017 is



25

Figure 8: Low Pass Filter GEOMAG software

shown in Figures 9a-9d (red color) for all the components. The despiked data is illustrated in Figures 9a-9d (red color) and is further illustrated in Figure 9e, selecting a small portion of the Z component dataset of the same day (highlighted with black

30  rectangular box).





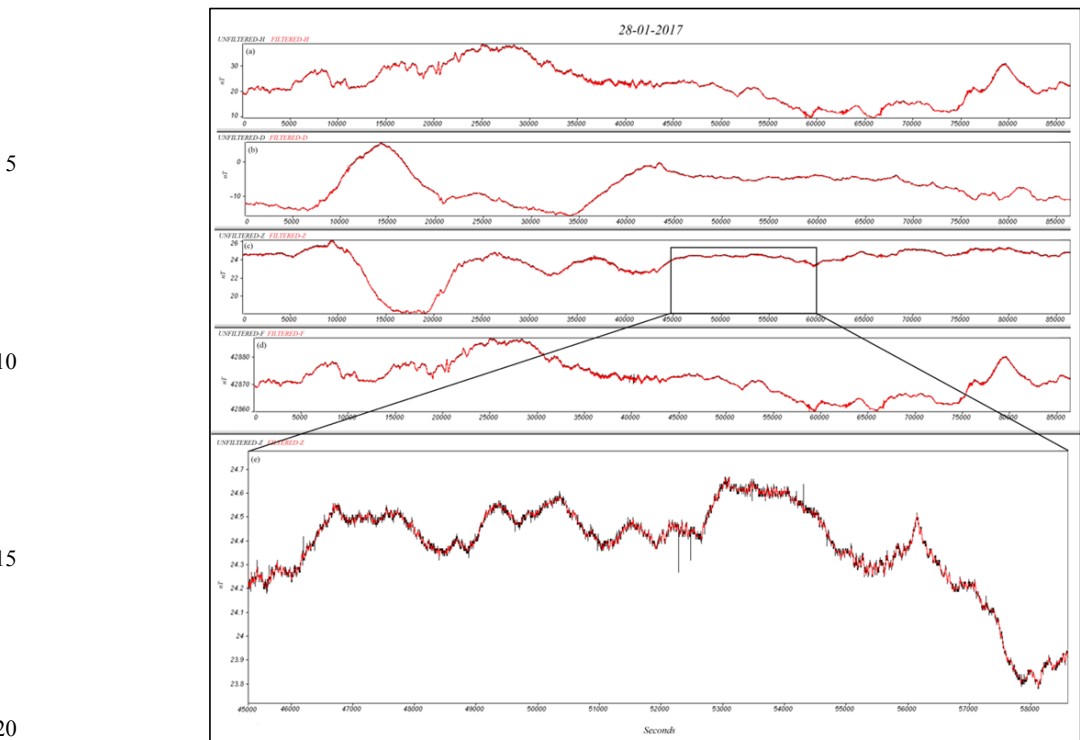

Figure 9: Sample 1 second raw and filtered data plots from the CPL Magnetic Observatory.

### 3.2 Raw and Filtered data sample from GEOMAG-02MO

25 The LPF GEOMAG software with different cut-off frequencies for a selected portion of observed spikes in the raw data file of Z component recorded on $27^{th}$ August, 2016, is shown in Figure 10a. This data file is directly uploaded to LPF GEOMAG software (the real-time data from GEOMAG-02MO creates a copy of the file in the hard disk of the PC apart from flash memory) installed on computer and the remaining procedures: filter window size and output data format follows the same explanation as explained in Figure 8, except by disabling the total field values and the changes in the cut off frequencies in

30 the software (middle panel, Figure 8). Figures 10b-10d shows the output of the filtered data of Z component with cutoff frequencies: 0.2 Hz, 0.1 Hz and 0.05 Hz. The LPF GEOMAG software is tested for many days at IMO-HYB before the permanent installation of GEOMAG-02MO at CPL Observatory.





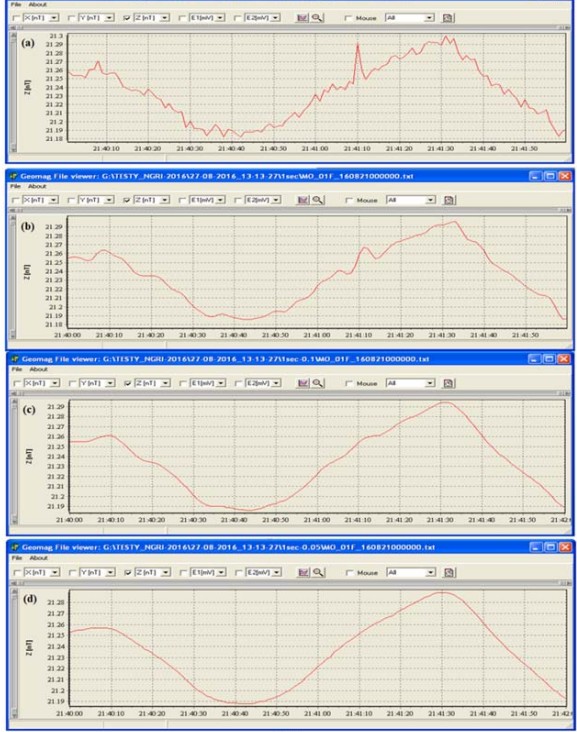

Figure 10: Despiking of data using Low Pass Filter GEOMAG software with different cut-off frequencies.

### 3.3 Effects of temperature and time stamping errors

The temperature sensitivity is one of the most important parameters affecting the long-term stability of magnetometers used in the Observatory. Stable temperature environment is required in the magnetometer hut to achieve high precision measurements. Attempts were made to test the temperature influence on the magnetic measurements at CPL. For this, the GEOMAG-02MO sensor and logger units installed in the newly constructed variometer vaults without any thermal insulation. The clear signature of temperature influence on the 1 sec daily variations is shown for 20[th] December 2016 as a sample (Figures 11a and 11b) and is more evident in ΔF variations (residual field between the GEOMAG-02MO and GSM90-F1) with respect to the recording system. Good correlation is observed between ΔF and variation in sensor vault temperature, shown in Figure 11c.Temperature variation of 2.5°C over a day is observed in


the sensor vault (blue color) and 0.8°C observed at logger vault (red color), which ensued 1.3 nT variation in ΔF over a day, Figure 11c. The changes in temperature are controlled by using EPS material, as described earlier. Figures 12a-12c shows 1 sec raw daily variations of H, D, Z, F and ΔF and variations from the recording system in a temperature controlled environment on one selected day (01.01.2017). It is evident from Figure 12c that the range of ΔF variations

5   over a day is 0.3 nT and the temperature variations in sensor and logger vaults are almost constant over a day in the range of 0.2°C.

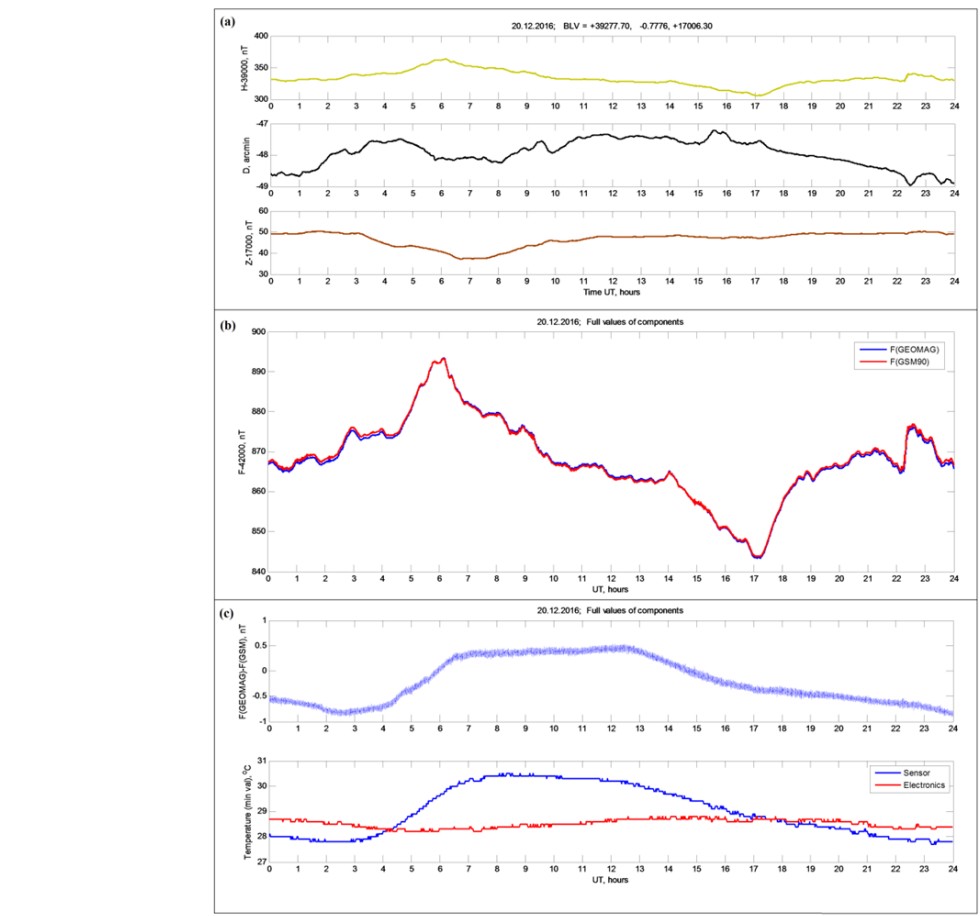

30   Figure 11: Example plot showing the influence of temperature on the field components (a) H, D and Z variations (b) Total field from GSM90-F1 and GEOMAG-02MO and (c) ΔF and Sensor-Logger temperatures observed on 20-12-2016, before specialized insulation; strong correlation between ΔF and Logger temperature.



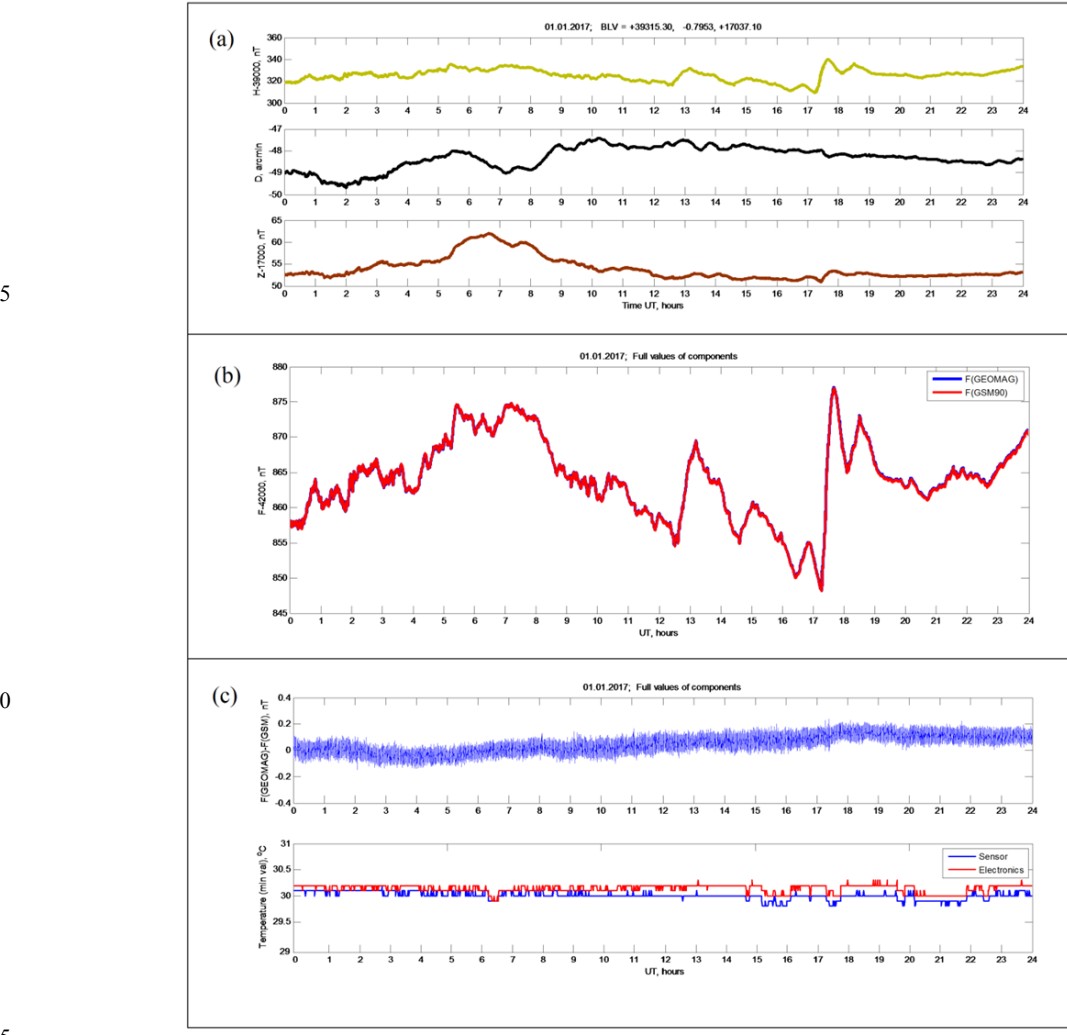

Figure 12: Example plot showing the influence of temperature on the field components (a) H, D and Z variations (b) Total field from GSM90-F1 and GEOMAG-02MO and (c) ΔF and Sensor-Logger temperatures observed on 01-01-2017, after specialized insulation.





Few milliseconds of delay noticed between the two recording units, GEOMAG-02MO and GSM90-F1, detected by the MAGREC-4B data acquisition system. After a few days of analysis on time stamping between the systems, the GSM90-F1 GPS is not exactly synchronizing with the GPS of the GEOMAG-02MO. The time stamping issue is resolved by upgrading

the software of the GSM90-F1 system and the corresponding changes made in the MAGREC-4B data acquisition system, to make sure about the time stamping differences between the systems is accurate.

**4 Baselines**

Two visits per week are scheduled for performing absolute observations for providing baselines to the variation data for the recording systems. Absolute measurements are regularly being performed twice a week at CPL Magnetic Observatory using MAG-01H Declination / Inclinometer system mounted on a non-magnetic pillar. A sample baseline for the month of Dec 2016 is shown in Figure 13. The performance of the recording systems is evaluated at IMO-HYB on daily basis by using the internet connectivity. In case of any emergency, the staff of IMO-HYB visit CPL apart from the scheduled tasks for

resolving the issues like, failure of internet, power, replacement of batteries, charge controllers, fuses and etc., to bring the situation to the normal.


Figure 13: Baselines of Dec 2016 using data from GEOMAG02-MO.




## 5 Conclusions

The generation of accurate and consistently good quality data over a long period of time from Observatories in low latitude regions continue to pose challenges. Socio-economic-political factors are often unfavorable for such activities. Under such circumstances it becomes more crucial to make utmost efforts for continuation and careful nurturing of such low latitude Observatories, which are already established so that observations of this region of the magnetic field are made possible. CPL is one such Observatory, which is located in low latitude, stable shield region, relatively devoid of anthropogenic EM sources. It has been possible to generate 1 second data in typical low latitude conditions with a combination of new instrumentation, software and acquisition techniques. Efforts over the past year have culminated in the production of continuous 1 second magnetic measurements, which can make vital contributions to the understanding of ULF waves, storm time magnetospheric ring current systems, sudden impulses and space weather phenomena.

## Acknowledgements

Authors would like to thank Director, CSIR-National Geophysical Research Institute and HEART project for the financial support towards procuring of the instruments and setting up of CPL, as well as permission to publish the work. This work was supported by the grants of DST-RFBR 16-55-45007.

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
