# Peer review of "One second vector and scalar magnetic measurements at low latitude observatory, CPL"

_Geoscientific Instrumentation, Methods and Data Systems, 2017_

## Referee Comment (RC1) · Anonymous Referee #1 · 19 Apr 2017

Review of 'One second vector and scalar magnetic measurements at low latitude observatory', CPL by Phani Chandrashekhar Nelpatla et al., submitted for publication in Geosci. Instrum. Method. Data Syst.

The manuscript describes the Chouttupal (CPL) geomagnetic observatory and its instruments. This is an important and laudable endeavour. I have, though, serious doubts if this manuscript really is contributing to the existing knowledge in our field or if it could be instructive to newcomers to the field. Therefore, I suggest mayor revisions and substantiate this suggestion by the following major and minor comments. I would like to encourage the authors to improve the manuscript considerably in form and content.

[Figure]

There are several misconceptions in the manuscript that already become clear in its abstract. Firstly, 1 second data is not 'in lieu of' (as claimed on page 1, line 15), but an addition to 1 minute data; the latter remains being the primary product of any IN-TERMAGNET observatory. Secondly, the authors claim that 1 second data is more compatible with measurements from LEO satellites (line 17) and contributes to the development of space weather forecasts (line 19). The authors do not give any evidence of these claims. I am not aware of examples that would support these claims and I had hoped the authors could educate us in this respect. Similarly, on page 2, line 16 to 19 the authors give examples for relevant science areas. However, all these science goals are easily achievable with standard equipment like FGE magnetometers.

What is the purpose of Fig. 5?

A major problem of the manuscript is that it presents technical data and screenshots of commercially available products. These figures and tables likely come together with these commercial products in the form of advertisement material or manuals. We learn very little about the source of these figures and the validity of the experimental procedures and analysis methods on which they are based (e.g. Figs 4, 6).

Also, it is not possible to evaluate the information collected and presented by the authors. For example, the temperature stability of their enclosure design can not be evaluated since only 1day of temperature data is presented. We wonder, how stable the temperature in the variometer enclosure is in the long tern, say during weeks or even a year. Judging from the baselines presented in Fig. 13 (changes of 5 nT in Z and 8 nT in H, D is illegible, so we don't know), the quality of the variometer or the temperature stability of its enclosure must be poor. Please present convincing temperature data over the course of several months or a year. Please present stable baselines if you want to advocate your mode of operations and the instruments you use.

Figures are illegible (Figs. 1, 6, 9, 10) or hardly legible (Figs. 11, 12, 13), invalidating any information that they might contain about the quality of the 1 second data presented

here.

Please mention the Narod magnetometers (page 4, line 7).

Please use SI units (e.g., page 4, line 19).

A variometer oriented to magnetic north should not compensate the magnetic field at its Y sensor (page 5, line 29), because standard methods of calibration will then fail to orient the horizontal sensors correctly. What precautions/methods have been used in this respect to guarantee correct calibration at CPL?

From reading the text, I don't understand the filter procedure described in paragraphs 2.2, 3.1 and 3.2. Looking at the figures helps little to understand the situation at CPL since the axes labels are too small to be readable and I don't know the time scales of Figs. 9 and 10. Please make clear what part of text is a general description of the commercial products on the one hand (and avoid writing too much about them, if possible) and what part of the text is describing the settings you chose for CPL (and please explain why they were chosen, how they were realised and what the results are). A cut off frequency of 0.005 Hz (200 seconds?) is mentioned in paragraph 3.1. Is this used in CPL and why was it chosen? The plot in Fig. 4 suggests that the signal is essentially zero at 0.2 Hz. My major concern here is that this is still part of the INTERMAGNET 1 second standard pass band and that the signal should not be altered here (a maximum gain/attenuation of 3 dB is foreseen in the pass band, see http://www.intermagnet.org/publications/im_tn_06_v1_0.pdf).

General remarks to the data processing: I would expect that the de-spiking is performed at a primary sampling rate (100 Hz), and not at the 1 Hz sampling rate. Also, my feeling is that the filtering used here is just smearing out spikes rather than removing them. Finally, I am in doubt about the use of the Magrec 4 data logger here. It looks like the time stamping and the filtering is already done by the variometer (which has a digital output), thus main features of the Magrec remain unused and it is questionable why using such a sophisticated data logger for the simple task of saving and
transferring data.

I don't see a proper analysis and discussion of socio-economic-political factors in the manuscript and would remove this part from he conclusion. Where there any low-latitude aspects in the manuscript that would justify the title of the paper?

The English needs considerable improvement.

---

## Referee Comment (RC2) · Anonymous Referee #2 · 17 May 2017

The article "One second vector and scalar magnetic measurements at low latitude observatory, CPL" presents a detailed description of the recent upgrades of CPL observatory site, including new magnetic sensors, data loggers and internet connectivity to ensure prompt and reliable 1-second magnetic data. It presents the data acquisition system and baseline operations. It is highly informative, but sometimes the text is not very clear and some part contain too many details, for instance on software description. I therefore recommend minor revisions. I include a list of points that should be addressed.

Some acronyms are not explicitly defined in the text (e.g. on page 1 line 28: CSIR; on page 7, line 2: MT, page 12, line 12: OFC ...)

[Figure]

page 1, line 24: I think that the BCMT yearly bulletin, that does not discuss the instruments developments for 1-second acquisitions, is not the more suitable citation here. I suggest to cite instead: Chulliat, A., J. Savary, K. Telali, and X. Lalanne (2009), Acquisition of 1-second data in ipgp magnetic observatories, in Proceedings of the XIIIth IAGA Workshop on Geomagnetic Observatory Instruments, Data Acquisition, and Processing, edited by J. J. Love, Open-File Report 2009–1226, pp. 54 – 59, U.S. Geological Survey.

page 1, line 31: for the GEOMAG-02MO sensor, the manufacturer was indicated, I suggest to do the same for the MAGREC-4B.

page 2, line 3: I suggest to add INTERMAGNET to the list of keywords.

page 2, line 8: I suggest to indicate explicitly that the data are Earth magnetic field data

page 2, lines 9-12: most of the institutes cited operating many observatories they do it in collaboration with other institutes. I suggest to modify this list (for instance USGS, GSC, GFZ, EOST are also contributing with a large number of observatories) or to remove it.

page 2, line 14: I prefer the use of the word "data" as plural word: I suggest to replace "is" with "are".

page 3, figure 1: the upper panel includes axes label indicating latitude and longitude values, but the numbers shown are constant on the small area shown. The lower panel contains many annotations that are written with a small font, that is not easily readable on print. I suggest to incread the fonts and use colours that are contrasting with the background image.

page 3 line 19: I suggest to revise the sentence to make it more straightforward: "The CPL observatory is located 60 km... and was developed ...".

page 3, lines 20-23: I suggest to revise the description of the observatory setup to give some more information aimed at readers not familiar with magnetic observatories.

page 4, lines 4-14: I think it would improve clarity to describe the history of CPL instrumentations, indicating briefly the reasons behind the installation of additional instruments etc.

page 5, figure 2: correct "Lightning" on the figure description. I again suggest to increase the fonts and make text more easy to read.

page 5, line 28: 'Y' component indicates the East (geographic) component of the magnetic field. I think here the recorded component should be indicated either as E or D.

page 6, figure 3b: The graph contains many details, but there is no indication about the GPS receivers used for time stamping. It could be useful to add this information.

page 6, line 18: I suggest to add the INTERMAGNET technical note in the Reference list, removing its title from this sentence.

page 6, line 19: I suggest to use the word "issues", instead of "problems"

page 7, figure 4: an indication about the experimental conditions when these graphs were computed, could be useful. Did they were acquired at CPL? How long was the acquisition?. . .

page 7, line 6: the text within brackets is reproduced form INTERMAGNET technical note, but it not easily understandable by a reader: I suggest to write "Phase response, maximum group delay").

page 10, line 9: I suggest to indicate the manufacturer of Magrec-4B.

page 10, line 26: I would suggest to use a different expression than "raw GPS data", since the position solution is not a raw GPS measurement. Raw GPS data would be the pseudorange and phase values of each satellite acquired, before computing a position solution.

page 11, figure 6: this figure contains too many screen-shots that are rendered in a

small space. I suggest to reorganise it to have more readable panels. I do not think all 9 panels are necessary for the publication.

page 12, line 10: I would suggest to use a different word than "establishment", e.g. "institute" or "campus".

page 12, line 14: since it was already indicated that the Hyderabad computer is a Windows machine, there is no need to repeat it.

page 13, lines 1-17: I think that this description provides too many operational details that are not needed for this publication.

page 15, figure 9, page 16, figure 10 and relevant text in the manuscript: I think that using low pass filter to despike the data is not the best approach, since it filters out also many other geophysical signals. It would be preferable to flag the data points that are affected by unwanted noise and remove them when producing quasi-definitive data by substituting their values with the "missing value" used in INTERMAGNET.

page 20, lines 4-12: I suggest to add some additional information to recall the upgrades of CPL observatory presented in the article.

page 21, line 27: the complete citation of Turbitt et al., 2012 should be: Turbitt, C.; Matzka, J.; Rasson, J.; St-Louis, B.; Stewart, D., An instrument performance and data quality standard for INTERMAGNET one-second data exchange. [Poster] In: XVth IAGA Workshop on Geomagnetic Observatory Instruments and Data Processing, Cadiz, Spain, 4-14 June 2012.

---

## Author Comment (AC1) · 22 Jun 2017

We sincerely thank the reviewer for constructive criticisms and valuable comments, which were of great help in revising the manuscript. Accordingly, the revised manuscript has been systematically improved with new information. Our responses to the referee's comments are given below.

Comment (1): Firstly, 1 second data is not 'in lieu of' (as claimed on page 1, line 15), but an addition to 1 minute data; the latter remains being the primary product of any INTERMAGNET observatory.

[Figure]

Reply: Corrected the sentence and included "in addition to the traditional 1 minute data" in text as suggested. Lines: 17-18.

Comment (2): Secondly, the authors claim that 1 second data is more compatible with measurements from LEO satellites (line 17) and contributes to the development of space weather forecasts (line 19). The authors do not give any evidence of these claims. I am not aware of examples that would support these claims and I had hoped the authors could educate us in this respect.

Reply: Corrected the language in the abstract and few parts of the text are rewritten. Provided evidences to the reported claims in the text. Lines: 20-27. Our intention in this paper is to highlight the possible studies by using the 1 sec data and the relevant claims were mentioned above in the current version.

There is increasing demand from the global geomagnetic community for the recording of 1 second vector and scalar magnetic data in addition to the traditional 1 minute data, which would be more useful for (i) understanding the global scale ultra-low frequency (ULF) waves and sudden impulses (Chulliat et al. 2009; Agapitov and Cheremnykh, 2013); (ii) synchronizing the ground observatory data with low-Earth orbiting satellites for the development of high resolution magnetic models (Matzka et al. 2010; Love and Chulliat, 2013; Stolle et al. 2016); (iii) development of real-time space weather applications using the geomagnetic pulsation indices (Nosé et al. 2012; Xu et al. 2013); (iv) studying the ionospheric magnetic fields, Sq field, equatorial electrojet and lower-middle atmospheric dynamics (Chulliat et al. 2016) and (v) investigations on high-latitude magnetospheric-ionospheric physics with magnetometer networks (i.e., IMAGE, CANOPUS, GMC, etc.), (Chulliat et al. 2009b).

Comment (3): Similarly, on page 2, line 16 to 19 the authors give examples for relevant science areas. However, all these science goals are easily achievable with standard equipment like FGE magnetometers.

Reply: All the data and our analysis are generated by FGE magnetometers. The

manuscript revolves around conditions of installation and transmission of data with respect to old and new models of FGE magnetometers.

Comment (4): What is the purpose of Fig. 5?

Reply: The relevance of Figure 5 is now added in the manuscript. Lines: 7-9, Page: 10.

The purpose of Figure 5 is to report the observed night time horizontal field variations from GEOMAG-02MO with the established 3-component Fluxgate Magnetometer (DTU Space, Denmark) by using Spectral decomposition technique. It is evident from Figure 5 that the dynamic spectra obtained from GEOMAG-02MO showing similar response of signal as observed in FGE, highlighting the variations recorded at GEOMAG-02MO are closely in agreement with FGE.

Comment (5): A major problem of the manuscript is that it presents technical data and screenshots of commercially available products. These figures and tables likely come together with these commercial products in the form of advertisement material or manuals.

Reply: The screenshots provided in the manuscript are from the commercial product, MAGREC-4B which was configured for the first time with the 3-component GEOMAG-02MO with GSM-90F1. To achieve this, standard script files were modified to accommodate the formats of the output data of a new FGE for optimum archival and display.

Comment (6): We learn very little about the source of these figures and the validity of the experimental procedures and analysis methods on which they are based (e.g. Figs 4, 6).

Reply: Figure 4 represents the basic frequency-noise response of the new GEOMAG-02MO FGE sensor, generated at controlled conditions at manufacturing location. The manuscript includes the description of the noise characteristics and the improvements made by Geomagnet, Ukriane, for bringing the Observatory grade 1 second magnetometer compared to earlier GEOMAG-02M (see Reda and Neska, 2016). The best technical specifications of the GEOMAG-02MO are also mentioned in Table-01.

The plots shown in Figure 6 illuminate the various stages of development of the new FGE installation and the parameters associated with it.

Comment (7): Also, it is not possible to evaluate the information collected and presented by the authors. For example, the temperature stability of their enclosure design cannot be evaluated since only 1day of temperature data is presented. We wonder how stable the temperature in the variometer enclosure is in the long term, say during weeks or even a year.

Reply: The manuscript centers around documenting the crucial initial stages of development and establishment of a Magnetic observatory expected to produce good quality of data in the long term. It is true that in some examples only a short duration of data has been presented as a representation of the evolving stages of improvements of the setup and thereby of the data. CPL observatory itself is barely 3 years old. The new FGE was commissioned just about a year earlier and the improvements during this time are being showcased in the manuscript. In the revised version we are providing one complete month of observed variations and the temperature stability of the enclosure design are shown in Figure 12. The Figure 12 of earlier submitted version is now replaced with the new figure and labeled with the same figure number.

Comment (8): Judging from the baselines presented in Fig. 13 (changes of 5 nT in Z and 8 nT in H, D is illegible, so we don't know), the quality of the variometer or the temperature stability of its enclosure must be poor. Please present convincing temperature data over the course of several months or a year. Please present stable baselines if you want to advocate your mode of operations and the instruments you use.

Reply: The baselines of December 2016 were admittedly affected by several interventions which were required for troubleshooting. Subsequently we have achieved better

quality baselines. An example of good quality and stable baselines of H, D and Z components are shown in Figure 13 for January 2017. Figure 13 of earlier version is now replaced with the new Figure and continues with the same number.

Comment (9): Figures are illegible (Figs. 1, 6, 9, 10) or hardly legible (Figs. 11, 12, 13), invalidating any information that they might contain about the quality of the 1 second data presented here.

Reply: In the revised version of manuscript:

(i) Increased the font size in Figures 01, 09, 10 and 11.

(ii) The last bottom panel was removed in Figure 06.

(iii) Figures 12 and 13 are replaced with new Figures. Comment (10): Please mention the Narod magnetometers (page 4, line 7).

Included in text as suggested. Lines: 22, Page 4.

Comment (11): Please use SI units (e.g., page 4, line 19).

Reply: Included "meters" and changed the values accordingly in the text.

Comment (12): A variometer oriented to magnetic north should not compensate the magnetic field at its Y sensor (page 5, line 29), because standard methods of calibration will then fail to orient the horizontal sensors correctly. What precautions/methods have been used in this respect to guarantee correct calibration at CPL?

Reply: GEOMAG-02MO sensor was installed initially keeping 'X' component close to magnetic north and 'Y' component close to Zero by leaving 'Y' in uncompensated mode. But in the text the word "Un" was missing. After reaching 'Y' close to zero, the 'X' and 'Z' total field values were noted and in the next step 'X' component of the sensor is oriented towards south direction keeping 'Y' close to zero and the corresponding total field values were noted. Similarly, the 'X' component of the sensor is oriented towards East and later in West direction, keeping ' X' close to zero and the

corresponding total field values were noted. All these exercises will ensure the similar values in the 'Z' component in all the four directions to achieve perfect verticality. In other means the horizontal position of the sensor is also leveled equally in all the four directions.

The adopted installation procedure for GEOMAG-02MO at CPL Magnetic Observatory is discussed in detail below:

Step-1: Total: X= +39448.7 nT Y= +00001.3 nT "X" component in North direction and "Y" component close to Zero Z= +16923.1 nT

Step-2: Total: X= -39463.2 nT Y= -00011.7 nT "X" component in South direction and "Y" component close to Zero Z= +16927.4 nT

Step-3: Total: X= -00000.5 nT Y= -39523.1 nT "X" component in East direction and "X" component close to Zero Z= +16937.6 nT

Step-4: Total: X= -00000.4 nT Y= +39545.7 nT "X" component in West direction and "X" component close to Zero Z= +16916.7 nT

Average of Z component in North-South direction = +16923.1 + 16927.4 = 16925.3 (a)
Average of Z component in East- West direction = +16937.6 + 16916.7 = 16927.2 (b)

Mean of Z component from (a) and (b) = 16926.2

Again we leveled the sensor in all four directions (N, S, E and W) to achieve the mean value (i.e. 16926.2 nT), to ensure the sensor is vertically leveled. Finally the sensor brought back to "X" is in North direction and "Y" close to zero leaving in uncompensated mode.

Comment (13): From reading the text, I don't understand the filter procedure described in paragraphs 2.2, 3.1 and 3.2. Looking at the figures helps little to understand the situation at CPL since the axes labels are too small to be readable and I don't know the time scales of Figs. 9 and 10. Please make clear what part of text is a general

description of the commercial products on the one hand (and avoid writing too much about them, if possible) and what part of the text is describing the settings you chose for CPL (and please explain why they were chosen, how they were realised and what the results are).

Reply: The Figures 09 and 10 have been improved in the current version. Paragraph 2.2 describes the noise characteristics of different components of GEOMAG-02MO and the solutions offered for the reduction of non-orthogonality of components of the newly available GEOMAG-02MO sensor. The details, which are not so far commercially available to the public domain, are being presented through this manuscript.

Sections 3.1 and 3.2 illustrate the examples of data acquisition and processing of the new FGE at CPL.

Comment (14): A cut off frequency of 0.005 Hz (200 seconds?) is mentioned in paragraph 3.1. Is this used in CPL and why was it chosen? The plot in Fig. 4 suggests that the signal is essentially zero at 0.2 Hz. My major concern here is that this is still part of the INTERMAGNET 1 second standard pass band and that the signal should not be altered here.

Reply: The cutoff frequency of 0.005 Hz is not the part of INTERMAGNET 1 second standard. But we made an attempt to see the differences in the noise removed in the data after applying the INTERMAGNET 1 second standard cutoff frequency of 0.2 Hz (Figure 09a) and 0.005 Hz (Figure 09b) for CPL Observatory. It is evident from Figure 09a that the noise in the Z component (last bottom panel) was not completely removed with cutoff frequency 0.2 Hz and still shows the clear signatures of noise in the data set. With the cutoff frequency of 0.005 Hz (Figure 09b, last bottom panel), the noise in the Z component was completely removed. An example showing various cutoff frequencies is illustrated in Figure 10. For low-latitude Observatories especially in India, the influence of temperature, increase in industrialization and habitation are the critical constraints for recording the good quality noise free data, for which more

stringent filtering may be required. Our intention is to investigate the optimum filter required for our data based on our processing efforts with various filters. The occasional noise in the data will be removed by applying different cutoff frequencies or flagging the data by removing the data and substituting their values with the "missing value" to INTERMAGNET.

Comment (15): General remarks to the data processing: I would expect that the despiking is performed at a primary sampling rate (100 Hz), and not at the 1 Hz sampling rate. Also, my feeling is that the filtering used here is just smearing out spikes rather than removing them.

Reply: The despiking is performed at 1 Hz sampling rate. In the revised version, two cut off frequencies 0.2 Hz and 0.005 Hz are shown as example for one day (i.e. 18/08/2017) in Figures 09a and 09b with regards to removal of spikes at low-latitude Observatory, CPL.

Comment (16): Finally, I am in doubt about the use of the Magrec 4 data logger here. It looks like the time stamping and the filtering is already done by the variometer (which has a digital output), thus main features of the Magrec remain unused and it is questionable why using such a sophisticated data logger for the simple task of saving and transferring data.

Reply: GEOMAG-02MO variometer doesn't offer filtering the data directly during the time of recording but connected to GPS for time stamping of the data. The filtering software is available to the community as an additional tool.

As discussed already, MAGREC-4B data logger is integrating the 3 component vector data from GEOMAG-02MO and Scalar data from GSM-90F1. Further, the INTERMAG-NET Gaussian filter applied on 1 second data for minute means in IAGA 2002 format. MAGREC-4B plays a vital role in storing the GSM90-F1 data, as it doesn't have any internal memory and also identifies the time stamping differences between the flux-gate magnetometers, if any. It also provides backup data for GEOMAG-02MO, sent /

received commands between the host PC and instruments, latest acquired data and their changes, short and long-term real-time plots of acquired and filtered data, logs of different events and errors of the recording systems and the information about the unexpected shutdown of the systems in any case.

Comment (17): I don't see a proper analysis and discussion of socio-economic-political factors in the manuscript and would remove this part from the conclusion. Where there any low-latitude aspects in the manuscript that would justify the title of the paper?

Reply: CPL is a Magnetic Observatory established in low latitude region of India, which implies that the amplitudes of Z component are relatively small, at the same time anthropogenic noise in Z component is large, combined effects of high temperature and humidity are significant. Socio-economic factors dictate that ideal conditions for operation of Magnetic Observatory are extremely difficult to achieve. Under such circumstances we have utilized natural non-magnetic material and low cost construction methods to attempt to produce 1 second magnetic data of international standards. The manuscript documents various aspects of our evolving efforts for the same. It is hoped that it may provide a framework for scientists facing similar issues.

The revised version of manuscript highlights the above mentioned points in the conclusion section.

Comment (18): The English needs considerable improvement

Reply: In the revised manuscript, the English is improved.

**** For your kind perusal, Pdf is attached herewith as a supplementary which contains the the text and figures together.

Please also note the supplement to this comment: http://www.geosci-instrum-method-data-syst-discuss.net/gi-2017-16/gi-2017-16-AC1-supplement.pdf

[Figure]

[Figure]

**Fig. 1.** Figure 9a. Sample plots of 1 second raw (black colour line) and filtered data (red colour line) with cut-off frequency 0.2 Hz from the CPL Magnetic Observatory.

[Figure]

**Fig. 2.** Figure 9b. Sample plots of 1 second raw (black colour line) and filtered data (red colour line) with cut-off frequency 0.005 Hz from the CPL Magnetic Observatory.

[Figure]

**Fig. 3.** Figure 10. Despiking of data using Low Pass Filter GEOMAG software with different cut-off frequencies

[Figure]

**Fig. 4.** Figure 12. One complete month (i.e. January 2017) of observed variations and the temperature stability of the enclosure design of CPL Magnetic Observatory

[Figure]

**Fig. 5.** Figure 13. Computed baselines of H, D and Z components of GEOMAG-2MO for January-2017

---

## Author Comment (AC2) · 22 Jun 2017

We sincerely thank the reviewer for constructive criticisms and valuable comments, which were of great help in revising the manuscript. Accordingly, the revised manuscript has been systematically improved with new information. Our responses to the referee's comments are given below.

Comment (1): Some acronyms are not explicitly defined in the text (e.g. on page 1 line 28: CSIR; on page 7, line 2: MT, page 12, line 12: OFC ...)

Reply: The acronyms (CSIR, NGRI, MO, OFC, IP, and WINSCP) are defined in the

text.

CSIR: Council of Scientific and Industrial Research NGRI: National Geophysical Research Institute MO: Magnetic Observatory OFC: Optical Fiber Cable IP: Internet Protocol WINSCP: Windows Secured Copy Protocol We would like to replace the acronym MT with MO in the current version.

Comment (2): Page 1, line 24: I think that the BCMT yearly bulletin, that does not discuss the instruments developments for 1-second acquisitions, is not the more suitable citation here. I suggest to cite instead: Chulliat, A., J. Savary, K. Telali, and X. Lalanne (2009), Acquisition of 1-second data in ipgp magnetic observatories, in Proceedings of the XIIIth IAGA Workshop on Geomagnetic Observatory Instruments, Data Acquisition, and Processing, edited by J. J. Love, Open-File Report 2009–1226, pp. 54 – 59, U.S. Geological Survey.

Reply: Cited the reference in the text and removed Courtillot and Chulliat, 2008 as suggested.

Comment (3): Page 1, line 31: for the GEOMAG-02MO sensor, the manufacturer was indicated, I suggest to do the same for the MAGREC-4B.

Reply: Included the manufacturer for MAGREC-4B in the text as suggested.

Comment (4): Page 2, line 3: I suggest to add INTERMAGNET to the list of keywords.

Reply: Included INTERMAGNET to the list of keywords as suggested.

Comment (5): Page 2, line 8: I suggest to indicate explicitly that the data are Earth magnetic field data

Reply: Included 'Earth's magnetic field' as suggested.

Comment (6): Page 2, lines 9-12: most of the institutes cited operating many observatories they do it in collaboration with other institutes. I suggest to modify this list (for instance USGS, GSC, GFZ, EOST are also contributing with a large number of

observatories) or to remove it.

Reply: Included the institutes as mentioned in the text and also producing the data under collaboration program.

Across the world, 200 Magnetic Observatories are in operation, of which 150 are IN-TERMAGNET Observatories (IMOs) recording high quality 1 minute vector and scalar Earth's magnetic field data. Many of the IMOs started recording and producing 1 sec magnetic field data, prominent ones being 11 Magnetic Observatories operating by Institut De Physique Du Globe De Paris (IPGP) (Courtillot and Chulliat, 2008); 3 Polish observatories maintained by Institute of Geophysics Polish Academy of Sciences (Reda and Neska, 2016); 4 observatories operating by the Japan Meteorological Agency (JMA) (Minamoto, 2013); 8 observatories of British Geological Survey (BGS) (Thompson, 2014), and some others operated by different academic and research institutes under collaboration, for example USGS (United States Geological Survey), GSC (Geological Survey of Canada), GFZ (Geoforschungszentrum) and EOST (Ecole et Observatorie des Sciences de la Terre) (see for more information, http://www.intermagnet.org).

Comment (7): Page 2, line 14: I prefer the use of the word "data" as plural word: I suggest to replace "is" with "are".

Reply: Corrected as suggested. Line: 28, Page: 2

Comment (8): Page 3, figure 1: the upper panel includes axes label indicating latitude and longitude values, but the numbers shown are constant on the small area shown. The lower panel contains many annotations that are written with a small font that is not easily readable on print. I suggest to increase the fonts and use colours that are contrasting with the background image.

Reply: Increased the font size in the figure, as well as the resolution in the revised version of the manuscript.

Comment (9): Page 3 line 19: I suggest to revise the sentence to make it more straight-forward: "The CPL observatory is located 60 km: : : and was developed : : :".

Reply: Corrected the text as suggested. Lines: 11-12, Page: 4.

Comment (10): Page 3, lines 20-23: I suggest to revise the description of the observatory setup to give some more information aimed at readers not familiar with magnetic observatories.

Reply:The detailed information about the observatory setup was already discussed in Arora et al. 2016.

Comment (11): Page 4, lines 4-14: I think it would improve clarity to describe the history of CPL instrumentations, indicating briefly the reasons behind the installation of additional instruments etc.

Reply: The CPL Magnetic Observatory consists of Tri-axial Digital Fluxgate Magnetometer (DTU, Denmark) and GSM90-F1 Overhauser as primary variometer setup. MAG-01H theodolite was used for performing absolute observations since the day Observatory was established. The Observatory grade 1 second GEOMAG-02MO was also installed as a secondary Magnetometer with another GSM90-F1 Overhauser in the same Observatory campus. The idea of installing the secondary magnetometer is to provide back up to the primary variometer system, in case of any issues.

Comment (12): Page 5, figure 2: correct "Lightning" on the figure description. I again suggest to increase the fonts and make text more easy to read.

Reply: Corrected the spelling of 'Lightning' in the Figure and also modified the font size in the figures.

Comment (13): Page 5, line 28: 'Y' component indicates the East (geographic) component of the magnetic field. I think here the recorded component should be indicated either as E or D.

Reply: Included Declination (D) as suggested. Lines: 13-14, Page: 6.

Comment (14): Page 6, figure 3b: The graph contains many details, but there is no indication about the GPS receivers used for time stamping. It could be useful to add this information.

Reply: Included the label 'GPS' in Figure 2 and provided details of GPS receivers in the revised version. Lines: 5-6, Page:6.

Comment (15): Page 6, line 18: I suggest to add the INTERMAGNET technical note in the Reference list, removing its title from this sentence.

Reply: Included the citation of INTERMAGNET technical note in the reference list and removed the title from the text.

Turbitt, C., Benoit St-Louis, Jean Rasson, Jurgen Matzka, Duff Stewart, Danish Technical University, Xavier Lalanne, Gerhard Schwarz, Tom Shanahan., INTERMAGNET Definitive One-second Data Standard, Document number: TN6 , Version number: v1.0, 2014.

Comment (16): Page 6, line 19: I suggest to use the word "issues", instead of "problems".

Reply: Replaced the word 'issues' in the text as suggested.

Comment (17): Page 7, figure 4: an indication about the experimental conditions when these graphs were computed could be useful. Did they were acquired at CPL? How long was the acquisition?

Reply: Figure 4 represents the basic frequency-noise response of the new FGE sensor, generated at controlled conditions at manufacturing location. The manuscript includes the description of the noise characteristics. The data was not acquired from CPL but the acquisition was made few months ago before the shipment of sensor to Hyderabad, India.

[Figure]

Comment (18): Page 7, line 6: the text within brackets is reproduced form INTERMAGNET technical note, but it not easily understandable by a reader: I suggest to write "Phase response, maximum group delay").

Reply: Corrected the text as suggested.

Comment (19): Page 10, line 9: I suggest to indicate the manufacturer of Magrec-4B.

Reply: Included the manufacturer of Magrec-4B in the text as suggested.

Comment (20): Page 10, line 26: I would suggest to use a different expression than "raw GPS data", since the position solution is not a raw GPS measurement. Raw GPS data would be the pseudorange and phase values of each satellite acquired, before computing a position solution.

Reply: In the current version the raw GPS data was replaced by GPS data.

Comment (21): Page 11, figure 6: this figure contains too many screen-shots that are rendered in a small space. I suggest to reorganize it to have more readable panels. I do not think all 9 panels are necessary for the publication.

Reply: The last panel was removed from the Figure. The new Figure is attached for reference. The appropriate text was also rearranged in the manuscript.

Comment (22): Page 12, line 10: I would suggest to use a different word than "establishment", e.g. "institute" or "campus".

Reply: Replaced the word "establishment" with "Institute" in the revised version.

Comment (23): Page 12, line 14: since it was already indicated that the Hyderabad computer is a Windows machine, there is no need to repeat it.

Reply: Removed "Windows machine" in the text as suggested.

Comment (24): Page 13, lines 1-17: I think that this description provides too many operational details that are not needed for this publication.

Reply: We wish to keep the description which might provide inputs to the new users, with regard to the data transmission from a remote location.

Comment (25): Page 15, figure 9, page 16, figure 10 and relevant text in the manuscript: I think that using low pass filter to despike the data is not the best approach, since it filters out also many other geophysical signals. It would be preferable to flag the data points that are affected by unwanted noise and remove them when producing quasi-definitive data by substituting their values with the "missing value" used in INTERMAGNET.

Reply: So far we are flagging the data points that are unwanted noise and removing them during the preparation of quasi-definitive data by substituting their values with the "missing value" to INTERMAGNET. We are not planning to despike the data by using the LPF software. The idea is to provide information about the filter software with two examples as shown in the Figures 09a and 09b and 10 .

Reply: The cutoff frequency of 0.005 Hz is not the part of INTERMAGNET 1 second standard. But we made an attempt to see the differences in the noise removed in the data after applying the INTERMAGNET 1 second standard cutoff frequency of 0.2 Hz (Figure 09a) and 0.005 Hz (Figure 09b) for CPL Observatory. It is evident from Figure 09a that the noise in the Z component (last bottom panel) was not completely removed with cutoff frequency 0.2 Hz and still shows the clear signatures of noise in the data set. With the cutoff frequency of 0.005 Hz (Figure 09b, last bottom panel), the noise in the Z component was completely removed. An example showing various cutoff frequencies is illustrated in Figure 10. For low-latitude Observatories especially in India, the influence of temperature, increase in industrialization and habitation are the critical constraints for recording the good quality noise free data, for which more stringent filtering may be required. Our intention is to investigate the optimum filter required for our data based on our processing efforts with various filters. The occasional noise in the data will be removed by applying different cutoff frequencies or flagging the data by removing the data and substituting their values with the "missing value" to
INTERMAGNET.

Comment (26): Page 20, lines 4-12: I suggest to add some additional information to recall the upgrades of CPL observatory presented in the article.

Reply: Hyderabad Magnetic Observatory, HYB (1964-present) and Choutuppal Geoelectric Observatory, CPL (1967-1991) were established with the intention of studying low-latitude magnetic phenomena at all frequency ranges. Operations at CPL were discontinued due to increased noise in electrical measurements. With rapid urbanization and introduction of Hyderabad Metro Rail project in the vicinity of IMO-HYB, it was necessary to re-look at possibilities of making noise free magnetic measurements in the erstwhile Geoelectric observatory at Choutuppal. Preliminary observations in 2012 and continuing observations thereafter, have led to recognition of Choutuppal (CPL) as a Magnetic Observatory by International Association of Geomagnetism and Aeronomy (IAGA). The quieter magnetic environment of the CPL campus with minimal human footprint along with carefully designed constructions to minimize effects of temperature fluctuations have led to improving qualities of data and baselines. CPL at present is under consideration of INTERMAGNET status. We have included these points in the revised version of manuscript.

It has been possible now to generate 1 second data in typical low-latitude conditions with a combination of new instrumentation, from the Observatory grade 1 second magnetometer and upgraded version of GSM-90 F1 together with established 3-component Fluxgate Magnetometer and Overhauser, software (Matlab based data processing) and acquisition techniques.

Comment (27): Page 21, line 27: the complete citation of Turbitt et al., 2012 should be: Turbitt, C.; Matzka, J.; Rasson, J.; St-Louis, B.; Stewart, D., An instrument performance and data quality standard for INTERMAGNET one-second data exchange. [Poster] In: XVth IAGA Workshop on Geomagnetic Observatory Instruments and Data Processing, Cadiz, Spain, 4-14 June 2012.
[Figure]

Reply: Included the complete citation in the reference list.

**** Figures 9, 10, 12 and 13 of earlier version were now modified and in the revised version the figures are attached for your kind reference and continues with the same figure numbers.

**** For your kind perusal, pdf is attached herewith as a supplementary which contains both text and figures together.

Please also note the supplement to this comment:
http://www.geosci-instrum-method-data-syst-discuss.net/gi-2017-16/gi-2017-16-AC2-supplement.pdf

———————————————————

**Fig. 1.** Figure 01. Bird's eye view of Choutuppal campus of CSIR-National Geophysical Research Institute (top panel) and the highlighted text box show the location of variometer vaults, Azimuth pillar, Absolut

[Figure]

**Fig. 2.** Figure 02. Thermally insulated new variometer vaults using Extruded Polystyrene (EPS) foam sheets for good temperature control and the location of sensor and logger vaults.

**Fig. 3.** Figure 06. Details of data logging and real-time plotting tools of MAGREC-4B data acquisition system

[Figure]

**Fig. 4.** Figure 9a. Sample plots of 1 second raw (black colour line) and filtered data (red colour line) with cut-off frequency 0.2 Hz from the CPL Magnetic Observatory.

[Figure]

[Figure]

**Fig. 6.** Figure 10. Despiking of data using Low Pass Filter GEOMAG software with different cut-off frequencies.

[Figure]

01-01-2017 to 31-01-2017

**Fig. 7.** Figure 12. One complete month (i.e. January 2017) of observed variations and the temperature stability of the enclosure design of CPL Magnetic Observatory

[Figure]

**(a)** H-Baseline

**(b)** D-Baseline

**(c)** Z-Baseline

**Fig. 8.** Figure 13. Computed baselines of H, D and Z components of GEOMAG-2MO for January-2017